# A Rapid Method for Sequencing Double-Stranded RNAs Purified from Yeasts and the Identification of a Potent K1 Killer Toxin Isolated from *Saccharomyces cerevisiae*

**DOI:** 10.3390/v11010070

**Published:** 2019-01-16

**Authors:** Angela M. Crabtree, Emily A. Kizer, Samuel S. Hunter, James T. Van Leuven, Daniel D. New, Matthew W. Fagnan, Paul A. Rowley

**Affiliations:** 1Department of Biological Sciences, University of Idaho, Moscow, ID 83844, USA; amcrabtree@uidaho.edu (A.M.C.); mrs@jkizer.com (E.A.K.); jvanleuven@uidaho.edu (J.T.V.L.); 2IBEST Genomics Core, University of Idaho, Moscow, ID 83843, USA; shunter@uidaho.edu (S.S.H.); dnew@uidaho.edu (D.D.N.); mfagnan@uidaho.edu (M.W.F.)

**Keywords:** mycovirus, dsRNA, sequencing, killer toxin, totivirus

## Abstract

Mycoviruses infect a large number of diverse fungal species, but considering their prevalence, relatively few high-quality genome sequences have been determined. Many mycoviruses have linear double-stranded RNA genomes, which makes it technically challenging to ascertain their nucleotide sequence using conventional sequencing methods. Different specialist methodologies have been developed for the extraction of double-stranded RNAs from fungi and the subsequent synthesis of cDNAs for cloning and sequencing. However, these methods are often labor-intensive, time-consuming, and can require several days to produce cDNAs from double-stranded RNAs. Here, we describe a comprehensive method for the rapid extraction and sequencing of dsRNAs derived from yeasts, using short-read next generation sequencing. This method optimizes the extraction of high-quality double-stranded RNAs from yeasts and 3′ polyadenylation for the initiation of cDNA synthesis for next-generation sequencing. We have used this method to determine the sequence of two mycoviruses and a double-stranded RNA satellite present within a single strain of the model yeast *Saccharomyces cerevisiae*. The quality and depth of coverage was sufficient to detect fixed and polymorphic mutations within viral populations extracted from a clonal yeast population. This method was also able to identify two fixed mutations within the alpha-domain of a variant K1 killer toxin encoded on a satellite double-stranded RNA. Relative to the canonical K1 toxin, these newly reported mutations increased the cytotoxicity of the K1 toxin against a specific species of yeast.

## 1. Introduction

Double-stranded RNAs (dsRNAs) found within fungi are the hallmark of infection by mycoviruses and their associated satellites. The majority of mycoviruses do not cause overt pathology in their host fungi, therefore, the direct extraction of dsRNAs and the visualization of viral particles using electron microscopy are the best methods to identify the presence of mycoviruses in fungal cultures. Surveys of pure fungal cultures indicate that dsRNA mycoviruses are abundant and present within every major group of fungi [1,2,3]. Although most mycoviruses appear to be avirulent, they can still lead to phenotypic changes in their fungal hosts, including changes in pigmentation, growth rate, and sporulation efficiency, and can improve stress tolerance, cause hypo- or hypervirulence in pathogens, or enable the production of extracellular antifungal toxins [4,5,6]. To better understand the contribution of mycoviruses to fungal ecology and pathogenicity, there is a need to improve the existing dsRNA purification and sequencing methods to enable the exploration of mycovirus diversity within fungi. 

The purification of dsRNAs from cell culture, tissues, and environmental samples has been achieved by applying a variety of techniques, including selective precipitation, solvent extraction, size exclusion chromatography, gel electrophoresis, affinity purification using dsRNA-binding proteins or cellulose, selective degradation of non-dsRNA nucleic acids, or a combination of these methods [7,8,9,10,11]. DNAs can be synthesized from a template of purified dsRNAs using reverse transcriptase, a reaction that requires priming to initiate synthesis in a 5′ to 3′ direction. The 3′ polyadenylation of cellular mRNAs can be used to prime cDNA synthesis using oligo(dT) primers, but the 3′ termini of linear mycoviral dsRNAs lack these terminal homopolymeric sequences. However, homopolymeric adenine-rich tracts that are known to be encoded internally by satellite dsRNAs found within yeasts have been successfully used to prime cDNA synthesis [12,13,14]. To initiate reverse transcription from an RNA template of unknown sequence, primer binding sites can be added to the 3′ termini, either by enzymatic polyadenylation [15,16,17,18,19] or the ligation of short oligonucleotides [11,20,21]. This enables the synthesis of cDNAs that are representative of the full-length dsRNAs. Alternatively, random hexamers can be used to generate cDNAs that are annealed and repaired to form double-stranded DNAs for cloning and sequencing [22,23]. 

Cloning of dsRNA-derived cDNA products into plasmids has been widely used to determine their nucleic acid sequence via Sanger sequencing. However, there has been a shift towards using next-generation sequencing (NGS) methods to determine the genetic sequence of dsRNAs and random amplification of cDNA ends (RACE) to resolve the terminal ends of dsRNAs. The use of NGS has also enabled the interrogation of pooled environmental samples, plant tissues, and animal-derived samples to identify dsRNA viruses, including those associated with disease [21,23,24,25,26]. However, there remain opportunities to improve the efficiency of dsRNA sequencing techniques by optimizing dsRNA purification, reverse transcription, and 3′ end tailing to increase the quality of cDNAs used for NGS sequencing. Here we report a comprehensive approach that leverages NGS technologies to determine the genetic sequence of dsRNAs purified from yeasts. Specifically, we combine rapid dsRNA extraction and cDNA synthesis protocols to create high-quality cDNAs for downstream NGS library preparation using an affordable ‘tagmentation’ procedure [27]. We demonstrate the optimization of 3′ polyadenylation of viral dsRNAs for ‘anchored’ oligo(dT) priming, which simplifies the reverse-transcription, amplification, and sequencing of cDNAs by NGS. This method has allowed the description of the diversity of dsRNAs found within a single strain of *Saccharomyces cerevisiae*.

## 2. Materials and Methods

### 2.1. Double-Stranded RNA Extraction

Double-stranded RNAs were extracted from both live yeast cultures inoculated in yeast peptone dextrose (YPD) broth and from commercial packets of dried yeasts (all yeast strains used in this study are described in Appendix A). YPD cultures were grown overnight at 30 °C and washed once with sterile water. Dried yeasts were rehydrated with 500 μL of sterile water and vortexed until homogenized. All cultures were then centrifuged for 5 min at 8000× *g* and the supernatant aspirated. For each extraction, approximately 0.04 g of wet biomass (approximately 1 × 10^9^ yeast cells) from YPD cultures and 0.06 g dry weight of dried yeasts was used for dsRNA extractions. The following protocol was modified from the dsRNA extraction method previously published by Okada et. al [28]. Cellulose columns were prepared by puncturing the bottom of a 0.6 mL tube with a hot 20-gauge needle and nesting it in a 2.0 mL tube. Approximately 0.06 g of cellulose powder D (Advantec, Japan) was added to the 0.6 mL tube, followed by 500 μL of wash buffer (1× STE (100 mM NaCl; 10 mM Tris–HCl, pH 8.0; 1 mM EDTA, pH 8.0) containing 16% (v/v) ethanol). Wash buffer was removed by a 10 s centrifugation, just before use. To extract dsRNAs, 450 μL of 2× LTE (500 mM LiCl; 20 mM Tris-HCl, pH 8.0; 30 mM EDTA, pH 8.0) containing 0.1% (v/v) beta-mercaptoethanol (14.3 M) (Amresco) was added to the harvested yeast cells. The cell mixture was vortexed for 3 min at 3000 rpm (Disruptor Genie, Scientific Industries, Bohemia, NY, USA). Fifty microliters of 10% (w/v) SDS solution and 500 μL of phenol–chloroform–isoamyl alcohol [25:24:1] pH 8.0 were added to the crude cell extracts and vortexed until homogenous. Samples were centrifuged at 20,000× *g* for 5 min and the supernatant was transferred to a clean tube and a second 500 μL of phenol–chloroform–isoamyl alcohol extraction was performed. A 0.2× volume of oligo d(T)_25_ magnetic beads (New England Biolabs, Ipswich, MA, USA) was added to the recovered supernatant before the sample was vortexed, agitated at 250 rpm at ambient temperature for 10 min, and then allowed to stand on a magnetic rack for 5 min. The supernatant was transferred to a clean tube whereupon a one-fifth volume of ethanol was added to precipitate the nucleic acids from solution. Tubes were centrifuged at 20,000× *g* for 3 min to remove precipitates and the supernatant was transferred to the pre-prepared cellulose spin column. The column was centrifuged at 10,000× *g* for 10 s, and the flow-through was discarded. Four hundred microliters of wash buffer was added to the columns, centrifuged at 10,000× *g* for 10 s and the flow-through was discarded. This step was repeated twice, for a total of three washes. After the last wash, the columns were dried by centrifugation at 10,000× *g* for 10 s. Cellulose columns were transferred to clean tubes, 400 μL of 1× STE was added, and columns were centrifuged at 10,000× *g* for 10 s to collect the eluate. Forty microliters of 3 M aqueous sodium acetate, pH 5.2, and 1 mL of absolute ethanol were added to the eluate, which was inverted to mix, and then centrifuged at 20,000× *g* for 5 min to precipitate the dsRNAs. The ethanol mix was aspirated, and dsRNA pellets were allowed to air-dry, before being suspended in 11 μL of nuclease-free water. To remove any remaining DNAs from the dsRNA-enriched sample, 0.5 μL of *E. coli* DNase I enzyme (New England Biolabs) was added with 1.2 μL of NEB Buffer 2.1, 0.5 μL of 10 mM CaCl_2_, and incubated at 37 °C, for 10 min. DMSO was added to a final concentration of 15% (v/v) and the sample was incubated at 95 °C, for 10 min to deactivate the DNase I and denature the dsRNAs, prior to cDNA synthesis. Samples were rapidly cooled in an ice bath to reduce the annealing of dsRNAs. A more rapid variation of this method for screening yeasts for the presence of dsRNAs was also used and involved only a single phenol:chloroform extraction, no oligo d(T)_25_ magnetic beads, and no DNase digestion. This rapid protocol gives higher yields and a clear visualization of the dsRNAs by agarose gel electrophoresis.

### 2.2. Sequencing Sample Preparation

Poly(A) polymerase (New England Biolabs) was used to synthesize a poly(A) tail at the 3′ termini of all denatured dsRNAs. To 12.5 μL of purified dsRNAs, the following was added: 1.5 μL 10× poly(A) polymerase reaction buffer, 1.5 μL adenosine 5′ triphosphate [10 mM], 0.5 μL of poly(A) polymerase (diluted 1:32 in nuclease-free water), and 0.5 μL murine RNAse inhibitor. Samples were incubated at 37 °C for 30 min, 65 °C for 20 min, 98 °C for 5 min, and then immediately placed in a wet ice slurry. Superscript IV (Invitrogen, Carlsbad, CA, USA) with an “anchored” NV(dT)_20_ primer (Invitrogen) was used to reverse transcribe the poly(A)-tailed single-stranded RNAs (ssRNAs) into cDNAs according to the manufacturer’s protocol. Murine RNase Inhibitor (New England Biolabs) was used in place of the RNaseOUT™ RNase Inhibitor. Each sample was digested with 1 μL of RNase H (New England Biolabs) and incubated at 37 °C, for 20 min to remove ssRNAs. cDNAs were annealed at 65 °C, for 2 h. To fully extend cDNA overhangs, 1 μL of *E. coli* DNA Polymerase I enzyme (New England Biolabs) was added to 3.5 μL of NEB Buffer 2.0 and 0.5 μL of 10 mM dNTPs and was incubated at 37 °C, for 30 min. DMSO was then added to a final concentration of 15% (v/v) and the reaction was incubated at 75 °C, for 20 min, to deactivate the polymerase. Five microliters of cDNAs were used as a template for PCR amplification, using 25 μL of Phusion Master Mix with HF Buffer (New England Biolabs), 1 μL of anchored oligo(dT) primer (0.7 ug/μL), and 1.5 μL of DMSO, to a final reaction volume of 50 μL. Reactions were subjected to the following parameters on a thermal cycler: (1) 72 °C for 10 min, (2) 98 °C for 30 s, (3) 98 °C for 5 s, (4) 50 °C for 10 s, and 72 °C for 45 s, (5) go to step 3 for 30 cycles, (6) 72 °C for 5 min. Six 50 μL PCR reactions were pooled and concentrated using HighPrep™ PCR reagent with magnetic beads, following the manufacturer’s protocol, using 0.5× sample volume of the reagent and five times the specified volume of ethanol wash (MagBio, Gaithersburg, MD, USA). Samples were eluted from the beads using 30 μL of nuclease-free water and subjected to fragment analysis (Fragment Analyzer, Advanced Analytical), prior to Illumina library preparation and NGS. 

### 2.3. Illumina Library Preparation Using a Modified Nextera Protocol

All cDNA samples were normalized to 2.5 ng/μL for the desired final average library insert size of 550 bp. Fluorometric quantification was performed with SpectraMax Gemini XPS plate reader (Molecular Devices, San Jose, CA, USA) and PicoGreen (Invitrogen). For the fluorometric quantification, 2 μL of cDNA was diluted in 98 μL 1× TE buffer (10 mM Tris-HCl, 1 mM EDTA, pH 7.5), and mixed with 100 μL of PicoGreen (diluted 1:200 in TE). Standards were prepared as per the manufacturer’s protocol and by scaling the volumes to one-tenth of that stated. Samples and standards were incubated at ambient temperature, in the dark, for 5 min, before analysis. Tagmentation, PCR (Applied Biosystems thermal cycler, Hercules, CA, USA), PCR-mediated adapter addition and library amplification were performed according to Baym et. al [27], with the post-tagmentation PCR using the following thermal cycling parameters: (1) 72 °C for 3 min, (2) 98 °C for 5 min, (3) 98 °C for 10 s, (4) 63 °C for 1 min, 72 °C for 30 s, (5) go to step 3 for 13 cycles, (6) 72 °C 5 min. For magnetic bead purification, 0.8× sample volume of HighPrep™ PCR reagent was used while following the manufacturer’s protocol. Samples were suspended in 50 μL of nuclease-free water and a two-sided size selection was performed to further narrow the insert size distribution. Then, 0.4× sample volume of HighPrep reagent was added to the sample with magnetic beads, and after an incubation at ambient temperature, for 5 min, the beads were discarded; 0.6× sample volume of HighPrep reagent was then added to the sample with magnetic beads and after incubation at ambient temperature for 5 min, the supernatant was removed. DNAs were then eluted from the magnetic beads and suspended in 50 μL of nuclease-free water. Samples were then quantified with a fluorometer and pooled by mass proportionally to the desired read distribution in the downstream sequencing run. Library-distribution, size-weighted fragment length, and nucleic acid concentration were determined by fragment analysis (Fragment Analyzer, Agilent Technologies Inc, La Jolla, CA, USA). 

### 2.4. Sequencing

The prepared DNA libraries were sequenced by the IBEST Genomics Resources Core at the University of Idaho, using an Illumina MiSeq sequencing platform and Micro v2 300 cycle reagent kit. Base calling and demultiplexing was performed using the Illumina bcl2fastq v2.17.1.14 software tool (Illumina, San Diego, CA, USA).

### 2.5. Bioinformatics Analysis

Bioinformatic analysis was done in two stages. First, to determine the approximate percentage of viral sequence, reads were mapped against a collection of previously published viral sequences using bowtie2 v 2.3.4.1 run with “--local” parameter [29]. Of the 471,742 reads sequenced for this sample, 97.84% could be mapped against viral sequences (NCBI GenBank accession numbers: ScV-L-A1, M28353.1; ScV-L-BC, NC_001641.1; ScV-M1, NC_001782.1). The resulting BAM file was further analyzed using SAMtools v1.5 to confirm the mapping depth across the full length of the viral reference sequence [30].

After confirming that the majority of sequenced reads were viral in origin, we performed a de novo assembly of reads in order to confirm the applicability of this method for the discovery of novel dsRNA viruses. Prior to assembly, HTStream (https://github.com/ibest/HTStream) was used to clean the reads. Due to the extremely high coverage, stringent cleaning parameters were used in order to retain the highest quality reads. Cleaning was done using the following steps and parameters: (1) PCR duplicates were identified and removed using hts_SuperDeduper; (2) reads were screened with hts_SeqScreener to remove PhiX control sequences, which were spiked in following Illumina protocols; (3) sequencing adapters were trimmed using hts_AdapterTrimmer; (4) reads were screened against a database of known sequencing adapters, using hts_SeqScreener and a collection of known adapter sequences to remove reads containing adapters that could not be trimmed during step 3; and (5) reads were quality trimmed by using a minimum q-score of 25 and retaining reads at least 148 bp in length using hts_QWindowTrim.

Cleaned reads were assembled de novo using the SPAdes assembler v3.11.1, with default parameters [31]. To assess the assembly quality and mapping depth, the contigs produced for each sample were used to build a bowtie2 index, and the cleaned reads from the respective sample were mapped. The resulting BAM files were visualized using Geneious 8.1 (https://www.geneious.com), which was also used to align the assembled contigs against previously published sequences for comparison. The read qualities were visualized in R using seqTools (R package version 1.14.0). Sequence reads were deposited to the NCBI Sequence Read Archive with the accession number: SAMN10274163. 

Polymorphic and fixed mutations were identified within the mapped reads, using Geneious v11.1.4. Significant mutations were selected from the output, using cutoffs for the minimum variant frequency (5 %) and minimum coverage (50 reads). Mutations with more than or equal to 95% variant frequency were specified as fixed, while the remaining mutations were considered polymorphic.

### 2.6. Cloning of dsRNAs

K1 toxin-encoding inducible plasmids were constructed by cloning reverse transcriptase PCR-derived K1 genes into pCR8 by TOPO-TA cloning (Thermo Fisher) using the primers PRUI1 and PRUI2 (Appendix A). The nucleic acid sequence of all cloned K1 genes was confirmed by Sanger sequencing. Utilizing Gateway™ technology (Thermo Fisher), K1 genes were sub-cloned into the destination vector pAG426-GAL-ccdB to create the high copy number, galactose-inducible plasmids pEK005 (reference K1 sequence) and pEK006 (K1 BJH001) [32]. To amplify and clone a putative polymorphic frameshift region from ScV-LA1, we used reverse transcriptase-PCR with primers PRUI132 and PRUI133. Amplified cDNAs were cloned into pCR8 by TOPO-TA cloning (Thermo Fisher) and the nucleic acid sequence was confirmed by Sanger sequencing.

### 2.7. Killer Toxin Assays

To test yeast strains for the production of killer toxins, single colonies were inoculated in YPD broth and grown at ambient temperature for 24 h. Putative killer yeasts were spotted at high cell density onto YPD dextrose ‘killer assay’ agar plates (0.003% w/v methylene blue, pH 4.6), seeded with a killer toxin-susceptible yeast strain. Plates were visually inspected for evidence of killer toxin production after incubation at ambient temperature, for 3 days. Toxin production by a strain of yeast was identified by either a zone of growth inhibition or methylene blue-staining of the yeasts that were spread as a lawn. To quantitatively compare the antifungal activities of the different K1 toxins, single colonies of *S. cerevisiae*, transformed with the plasmids pEK005 or pEK006, were inoculated in 1 mL of complete liquid media lacking uracil with 2% galactose. These cultures were incubated at ambient temperature, for 48 h, with shaking at 250 rpm. K1 toxin-susceptible yeasts were inoculated in 1 mL of YPD and incubated at ambient temperature for 48 h with shaking at 250 rpm. 6 × 10^5^ K1 toxin-susceptible yeast cells were spread onto YPD galactose killer assay agar plates (10% w/v galactose, 0.003% w/v methylene blue, pH 4.6). Five microliters with 6 × 10^6^ cells of K1-expressing yeast were spotted onto the inoculated plates and incubated at ambient temperature for 4 days. Areas of growth inhibition were determined by measuring the diameter of the growth inhibition zones. 

### 2.8. Verifying the Presence of DsRNA Elements in S. cerevisiae BJH001 by Reverse Transcriptase-PCR

DsRNAs extracted from *S. cerevisiae* BJH001 were used as templates for Superscript IV two-step reverse transcriptase-PCR, according to the manufacturer’s protocol, with primers specific for ScV-L-A1, ScV-L-BC, and ScV-M1 (Appendix A). 

## 3. Results

### 3.1. Extraction of High-Quality dsRNAs from Saccharomyces Yeasts

The presence of dsRNA mycoviruses in *S. cerevisiae* is often correlated with the production of antifungal proteins (killer toxins). We used two yeast strains that have been previously reported as killer yeasts (*S. cerevisiae* BJH001 [33] and *S. paradoxus* Y8.5 [34,35]), one non-killer yeast (*S. paradoxus* CBS12357), and several commercially-available dried yeasts to assay the effectiveness of a modified protocol based on a dsRNA extraction method previously optimized for filamentous fungi and plant material (Figure 1) [28]. Approximately 0.04 g of biomass (~1 × 10^9^ yeast cells) were used as input for the extraction of dsRNAs. Cells were first subjected to homogenization in LTE buffer, followed by two rounds of phenol:chloroform:isoamyl alcohol extraction. The resulting aqueous phase was then incubated with oligo(dT) beads to deplete cellular polyadenylated single-stranded RNAs (ssRNAs), before loading onto a cellulose D spin column. Eluted material from the cellulose D had a higher concentration of dsRNAs compared to a rapid method using guanidinium thiocyanate and phenol that we previously described (Appendix A) [33]. To remove residual DNAs, samples were incubated with DNase I. This protocol was used to identify the dsRNA content of several strains of *S. cerevisiae* “killer yeasts” that produce killer toxins (Figure 1A), which is often dependent on the presence of dsRNA totiviruses and associated satellite dsRNAs (Figure 1B) [5,36]. After extracting dsRNAs directly from rehydrated commercial dried yeasts or from yeasts grown in a laboratory culture, we were able to resolve dsRNAs in killer and non-killer yeasts that correspond to the presence of mycoviruses and satellite dsRNAs (Figure 1B). 

### 3.2. Next Generation Sequencing of cDNAs Derived from dsRNAs

To initiate reverse transcription and create full-length cDNAs from a purified mixture of dsRNAs, we used poly(A) polymerase to polyadenylate the 3′ end of denatured dsRNAs (Figure 2A). Analysis of dsRNAs, before and after poly(A) polymerase incubation by fragment analysis, revealed a significant increase in the molecular weight of the treated RNAs (Figure 2B). Reverse transcription was primed using an anchored oligo(dT) primer (sequence: NV(T_20_)) to minimize priming within the poly(A) tail. The resultant cDNAs were annealed and repaired by *E. coli* polymerase I and amplified with anchored oligo(dT) primers, using Phusion polymerase. After magnetic bead purification, the size distribution and quantity of cDNAs was determined by fragment analysis. Total cDNA yields ranged from 420–810 ng and had a broad size distribution (Figure 2C). The small size of mycovirus dsRNAs means that many different cDNAs can be analyzed using a fraction of the reads available during the NGS, therefore, the cost of conventional library preparation becomes a limiting factor for the sequencing of large numbers of fungal dsRNAs. To reduce the amount of time and resources required for the NGS library preparation, we applied a previously described inexpensive transposon-based ‘tagmentation’ method for preparing fragmented and tagged DNA libraries [27]. The resulting cDNA libraries were sequenced with an Illumina sequencing platform, using the MiSeq Sequencing v2 (Micro 300) package. Reads were cleaned, deduplicated, and trimmed, as described in the materials and methods. 

We found that high concentrations of poly(A) polymerase reduced the number of high-quality reads of viral origin after NGS and resulted in a large percentage of homopolymeric reads (Table 1). Titration of poly(A) polymerase was able to increase the overall number and quality of reads (25 U; 23,000 reads, 1.25 U; 28,000 reads, and 0.5 U; 42,000 reads), but we observed that using 0.02 U of poly(A) polymerase with an anchored oligo(dT) primer increased read count by 21-fold and reduced homopolymers by more than a 100-fold, relative to 25 U and a homopolymeric oligo(dT) primer (Table 1). In concert with the reduction in homopolymeric reads, we also observed an improvement in sequenced read quality. This improvement was caused by an increased base diversity and we were able to assemble long contigs of viral origin with a mean coverage of 610 (Table 1) (Figure 3A). This demonstrated that the enzymatic addition of the 3′ poly(A) tracts, which was previously used for the direct cloning of dsRNAs, is a feasible and rapid approach for the creation and NGS of dsRNA-derived cDNAs.

Using the SPAdes assembler, a de novo assembly of high-quality sequence reads derived from the dsRNAs extracted from *S. cerevisiae* BJH001, produced four long contigs with a high sequence coverage (Figure 3B). The most significant hits from the BLAST analysis of these contigs revealed that BJH001 harbors three distinct dsRNA species - two totiviruses (Saccharomyces cerevisiae virus L-A1 (ScV-L-A1) and Saccharomyces cerevisiae virus L-BC (ScV-L-BC)) and one satellite dsRNA (Saccharomyces cerevisiae satellite M1 (ScV-M1)) (Figure 3B). We have previously described the presence of ScV-L-A1 and ScV-M1 within this strain but were unaware of the totivirus ScV-L-BC [33]. Reverse transcriptase PCR was used to confirm the presence of these dsRNAs within the strain BJH001 (Figure 3B, inset). 

The assembly of the sequence reads onto the published reference sequences of these dsRNA viruses and satellite demonstrated that 99.9%, 99.5%, and 89.2% of the ScV-L-A1, ScV-L-BC, and ScV-M1 dsRNAs, were sequenced to a read depth greater than 50, respectively (Figure 3C). Median read depth for all dsRNAs was greater than 2500 (Figure 3C). The 3′ terminal ends of the dsRNAs were also resolved but with a low coverage (< 50 reads), especially for the terminal nucleotide (Appendix A). The 5′ terminal end of ScV-L-A1 and ScV-L-BC were also resolved at a low coverage, but we were unable to resolve the 5′ terminal nucleotide of the ScV-M1 (Appendix A). The only other region that was not well resolved was the low complexity ~200 bp homopolymeric adenine-rich tract contained within the ScV-M1 satellite dsRNA, which was masked prior to the read mapping. The increased coverage of the 5′ half of ScV-M1 was likely due to the initiation of the reverse transcription and PCR from this internal adenine-rich tract (Figure 3C) [14]. The overall high-quality and deep coverage of the dsRNAs present within the *S. cerevisiae* strain BJH001 using short-read Illumina sequencing, demonstrated the utility of the described method for the future discovery and characterization of novel mycoviruses. 

### 3.3. Sequence Variation in dsRNAs Identified by NGS

The high median read depth of our NGS datasets enabled the detection of fixed synonymous and non-synonymous mutations and indels within the dsRNAs extracted from the strain BJH001 (Figure 4) (Appendix A). Even though the dsRNAs isolated from the strain were extracted from a clonal population, single nucleotide polymorphisms and polymorphic indels were detected within both the ScV-L-A1 and ScV-L-BC contigs (Table 2). No polymorphic nucleotides were found in assembled contigs for the ScV-M1 dsRNA. Two polymorphic indels that are present together in 21% of the ScV-L-A1 dsRNAs caused a +1 frameshift followed, after 55 base pairs, by a -1 frameshift. However, we were unable to confirm these by reverse transcriptase-PCR, cloning, and Sanger sequencing, meaning that they could have appeared due to replication errors during sample preparation. The proximity of the observed mutations to the secondary structure of the frameshift region could account for the observed discrepancy. However, two fixed indels (one nucleotide insertion and one deletion) that were observed in all sequence reads of the ScV-L-BC, resulted in a small 4 amino acid frameshift within the C-terminus of the Gag-Pol fusion protein (Figure 4B). 

The *S. cerevisiae* strain BJH001 expresses a potent K1 killer toxin, which we have found to be capable of inhibiting the growth in a variety of different strains and species of yeast, unlike the non-killer *S. cerevisiae* strain BY4741 (Appendix A). Our NGS data suggests that the BJH001 K1 killer toxin differs from the canonical killer toxin gene sequence by two synonymous and two non-synonymous mutations (Figure 4C,D). To confirm the presence of the four mutations identified by NGS and test the functional significance of these mutations, we used reverse transcriptase PCR to directly amplify the K1 gene from the dsRNAs isolated from the strain BJH001. As a positive control, we also amplified the canonical K1 gene from the plasmid pM1TF (+) GAL [37]. The PCR products were cloned using TOPO-TA and Gateway™ methods into a galactose inducible yeast expression vector [32]. Importantly, the four mutations identified within the BJH001 K1 gene by Illumina NGS were confirmed by Sanger sequencing. The K1 expression vectors were used to transform the non-killer *S. cerevisiae* strain BY4741. To compare the biological activities of the two cloned K1 toxins, 6 × 10^6^ cells of each isogenic K1-expressing strain were spotted, in triplicate, onto galactose-containing agar plates seeded with various K1-sensitive yeasts. Qualitative comparison of the specificity of the BJH001 K1 toxin expressed from a plasmid or the dsRNA satellite demonstrated that the ectopic expression does not alter its specificity toward the toxin-sensitive yeasts (Appendix A). However, measurement of the area of growth inhibition revealed that the BJH001 K1 toxin produces significantly larger zones of growth inhibition on the K1-sensitive yeast *Kazachstania africana*, compared to the canonical K1 reference toxin (T-test, two-tailed, *p* < 0.01) (Figure 5). This zone of growth inhibition was 28% larger than the K1 reference toxin (Figure 5). For the other seven K1-sensitive strains tested, the differences between the two killer toxins did not significantly alter the area of the zone of growth inhibition, which suggests that the mutations in the K1 toxin from the strain BJH001 did not affect the amount of killer toxin produced or the rate of diffusion through the agar (Appendix A). These data demonstrated that mutations in K1 killer toxin can alter their toxicity to specific species of yeasts.

## 4. Discussion

The methods that we describe constitute a broadly applicable approach to the sequencing of dsRNA purified from fungi using Illumina NGS. We have successfully applied this approach to determine the nucleotide sequence of dsRNAs purified from the yeast *S. cerevisiae*. We also show the feasibility of extracting high-quality dsRNAs from commercial dried yeasts as well as laboratory grown cultures. Polyadenylation of dsRNAs was one of the first methods used to modify the 3′ termini of RNAs to enable cDNA synthesis for Sanger sequencing [15,16,18]. However, these methods were limited in their ability to clone full-length cDNAs derived from viral dsRNAs [9,18,38]. Most recent methods have focused on using 3′ oligo ligation or random priming to initiate cDNA synthesis from unknown dsRNAs, prior to cloning or NGS, and have been successful in determining the genetic sequence of many viral and satellite dsRNAs [11,12,13,19,20,21,35]. However, these previously described methods often involve labor- and time-intensive steps during dsRNA purification, 3′ oligomer ligation, and NGS library preparation. We have evaluated these methods to develop a protocol that is rapid and feasible for sequencing large numbers of small dsRNA molecules extracted from fungi. From cells to sequencer-ready libraries the described protocol takes 17 h; 4 h for dsRNA extraction, 8 h for cDNA synthesis, and 5 h for library creation by tagmentation [27]. CDNA synthesis and library construction using the described method takes 13 h, which compares well to contemporary commercial kits using mRNAs that take 12 h (TruSeq RNA sequencing kit; Illumina) and is faster than methods that require a long (up to 18 h) incubation for the efficient 3′ ligation of oligonucleotides, prior to cDNA synthesis [11,20,21]. To the best of our knowledge, the combination of 3′ polyadenylation, anchored oligo(dT) priming, and tagmentation for NGS library preparation has never been applied as a technique for the rapid synthesis of high-quality cDNAs from dsRNAs for Illumina NGS. 

Cloning of the dsRNA-derived cDNAs and 5′ or 3′ RACE enable the efficient resolution of the dsRNA terminal ends. NGS methods alone have been largely unsuccessful in the resolution of the terminal ends of cDNAs [21,39]. However, there are some notable exceptions that have leveraged a combination of commercial kits and homopolymeric primers to completely sequence the dsRNAs isolated from yeasts [12,13]. 

Our NGS method was also able to resolve most terminal ends when mapping to a reference sequence, but coverage appeared to be dependent on the terminal sequence of the dsRNAs. Moreover, we were able to resolve the 3′ termini of all dsRNAs within *S. cerevisiae* BJH001 but the 5′ termini had a reduced coverage (Appendix A). ScV-M1 and ScV-L-A1 have A/U-rich 5′ termini that may have resulted in ambiguities during sequence mapping, causing a poor sequence coverage. Except for the low coverage of the terminal nucleotides, we were able to assemble long viral contigs from the mixtures of different dsRNAs extracted from a single strain of yeast, independent of a reference sequence (Figure 3B). For example, we have previously studied the mycoviruses and dsRNAs present within *S. cerevisiae* BJH001 by agarose gel electrophoresis but were unaware of the presence of a variant ScV-L-BC within this strain because of its similar electrophoretic mobility to ScV-L-A1 [33]. By applying our NGS method we were able to identify the presence of ScV-L-BC and assemble a large contig with a high similarity to the reference sequence of this totivirus (Figure 3). The high coverage of the majority of the dsRNAs allowed the identification of fixed and polymorphic mutations within the populations of dsRNAs. We also observed an indel that resulted in a small, but dramatic change to the amino acid sequence of the polymerase domain in ScV-L-BC. This frameshifted region is peripheral to the conserved motifs of the catalytic core of the RNA-dependent RNA polymerase [40], which would suggest that these mutations might not disrupt the polymerase function. Furthermore, because ScV-L-BC is stably associated with *S. cerevisiae* BJH001, we do not expect that the fixed 4 amino acid frameshift to significantly affect viral replication and persistence. Frameshift mutations most often result in premature stop codons and defective proteins that are truncated, with the most prominent examples of frameshift mutations being those that cause human disease [41,42,43]. Moreover, mutant polymerases could be incorporated into viral capsids to form defective interfering particles with likely negative consequences for viral replication. Alternatively, frameshift mutations can result in novel protein functions, although they are less frequently reported [44]. The co-occurrence of a +1 and -1 frameshift indel suggests that there has been selection to maintain the reading frame of the polymerase gene, but any functional consequence to the polymerase enzyme and on the replication and persistence of ScV-L-BC remains unexplored. 

ScV-M1 was found to contain four fixed mutations in the K1 killer toxin gene that we were able to confirm by cloning and Sanger sequencing. The two non-synonymous mutations (I103S and T146I) map to the K1 alpha-domain that is important for the cytotoxicity of K1 and are positioned close to known mutations that are defective in cell wall binding and toxicity (D101R and D140R) (Figure 4D) [45]. Relative to the cloned canonical K1 toxin, the mutations I103S and T146I significantly increased the toxicity of K1 to *K. africana* but not to other strains of *Saccharomyces* yeasts that were challenged by the toxin (Figure 5). Previously, two different K1 variants have been described by reverse transcriptase PCR and Sanger sequencing within different species of *Saccharomyces* yeasts [34]. Expression of these variant K1 toxins in *S. cerevisiae* and *S. paradoxus* appeared to show that a single gain-of-function mutation in the K1 beta-domain (L251F) can increase the cytotoxicity of the K1 toxin, but the results were not quantified to assess the statistical significance [34]. Furthermore, a separate study failed to identify these K1 variants in the same strains of yeast [35]. We anticipate that our NGS method could be applied to rapidly elucidate the genetic sequence of satellite dsRNAs to investigate the effect of genetic variation on killer toxin activity. The large number of killer yeasts with unique antifungal activities discovered since the 1970s suggests that killer toxins are numerous and diverse [46,47,48,49,50,51,52]. Indeed, this is highlighted by the recent description of several novel satellite dsRNAs and associated killer toxins within *Saccharomyces* yeasts [35]. Ultimately, a better understanding of the relationship between killer toxin genotype and phenotype will clarify their contribution to fungal ecology with broad significance to human health and agriculture.

## Figures and Tables

**Figure 1 viruses-11-00070-f001:**
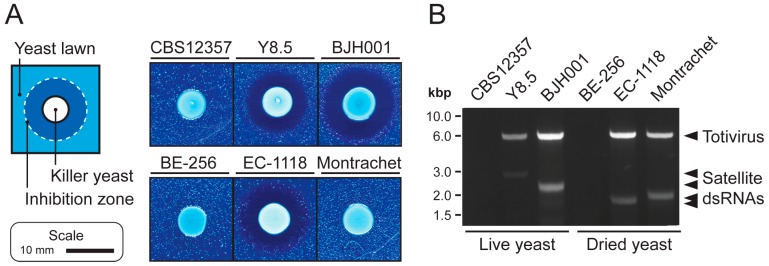
The extraction of dsRNAs from dried and actively growing cultures of killer and non-killer Saccharomyces yeasts. (**A**) Killer toxin production by different strains of *Saccharomyces* yeasts. The ability of yeasts to inhibit the growth of the lawn strain *Saccharomyces bayanus* CBS7001 indicate killer toxin production by strains Y8.5, BJH001, and EC-1118. (**B**) dsRNAs were separated by 0.8% agarose gel electrophoresis and stained with ethidium bromide. Molecular weight standards are DNA-specific and provide an approximate size of dsRNAs. Larger dsRNAs represent putative totiviruses, whereas the smaller heterogenous dsRNAs are strain-specific satellite dsRNAs.

**Figure 2 viruses-11-00070-f002:**
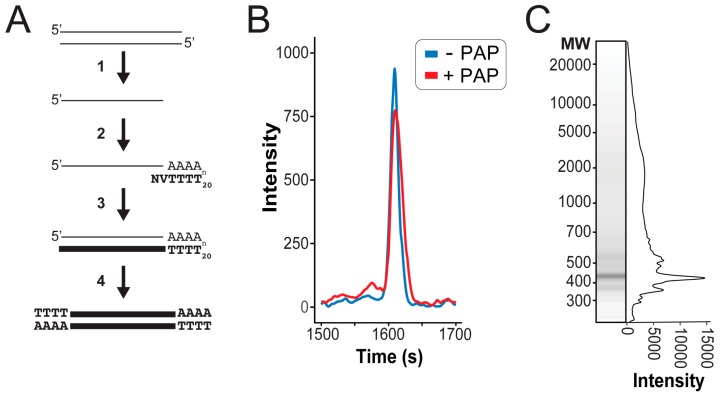
Overview of cDNAs synthesis from dsRNAs using poly(A) polymerase for next-generation sequencing. (**A**) (1) Purified dsRNAs are denatured and rapidly cooled to separate RNA strands. (**2**) ssRNA is 3′ polyadenylated by poly(A) polymerase and anchored oligo(dT) primers are annealed (N = any nucleotide, V = A, G, C). (**3**) Reverse transcription is initiated from anchored oligo(dT) primers to create cDNAs that are complementary to both the positive and negative strand of the dsRNAs. (**4**) RNAs are removed by RNase H digestion followed by the annealing and repairing of cDNAs. (**B**) The increase in the molecular weight of dsRNAs by the 3′ addition of adenine nucleotides by poly(A) polymerase (PAP) as visualized by intensity trace. (**C**) High molecular weight cDNA synthesis from dsRNAs extracted from *S. cerevisiae* by anchored oligo(dT) priming and PCR amplification visualized by intensity trace.

**Figure 3 viruses-11-00070-f003:**
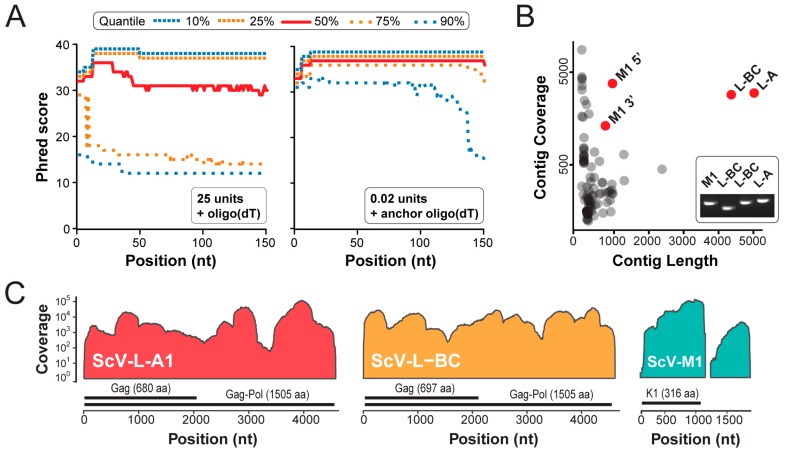
Next-generation sequencing of totiviruses and dsRNA satellites from *Saccharomyces cerevisiae* strain BJH001. (**A**) Read quality (phred score) after Illumina QC shown along the length of the sequencing reads when using different concentrations of poly(A) polymerase and oligo(dT) or anchored oligo(dT) primers; 10%, 25%, 75%, and 90% quantile and median (50% quantile) read quality at each position along the reads are shown. (**B**) Sequence contigs after de novo assembled represented by contig coverage and contig length. BLAST analysis of the four contigs with the longest length and deepest coverage enabled their identification as totiviruses (ScV-L-A1 and ScV-L-BC) and a dsRNA satellite (ScV-M1), the latter was assembled as two separate contigs. *Inset* reverse transcriptase-PCR was used to confirm the presence of each type of dsRNA. Two primer pairs were used to amplify the ScV-L-BC. (**C**) Read depth coverage across the reference-assembled ScV-L-A, ScV-L-BC, and ScV-M1 contigs. Open reading frames present within each dsRNA are shown above the nucleotide position.

**Figure 4 viruses-11-00070-f004:**
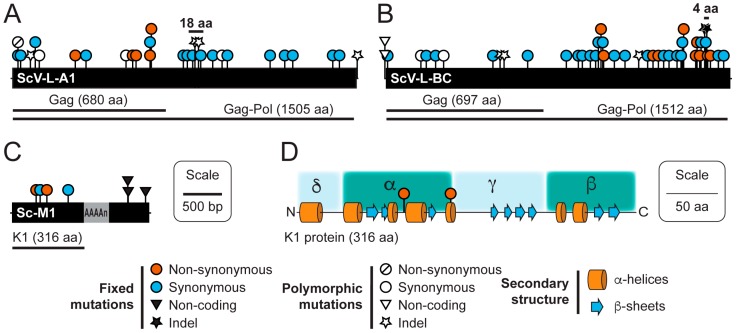
Natural variation in dsRNAs isolated from *S. cerevisiae* detected by NGS. A linear representation of the three dsRNAs isolated from the strain BJH001 showing the position of ORFs, relative to observed mutations (**A**) ScV-L-A1; 4575 bp, (**B**) ScV-L-BC; 4633 bp, and (**C**) ScV-M1; ~1700 bp. Bars above the indels represent the indel pairs that result in frameshifts of 18 and 4 amino acids in the Gag-Pol fusion protein. (**D**) Secondary structure prediction of the K1 killer toxin showing the position of the non-synonymous mutations in the strain BJH001.

**Figure 5 viruses-11-00070-f005:**
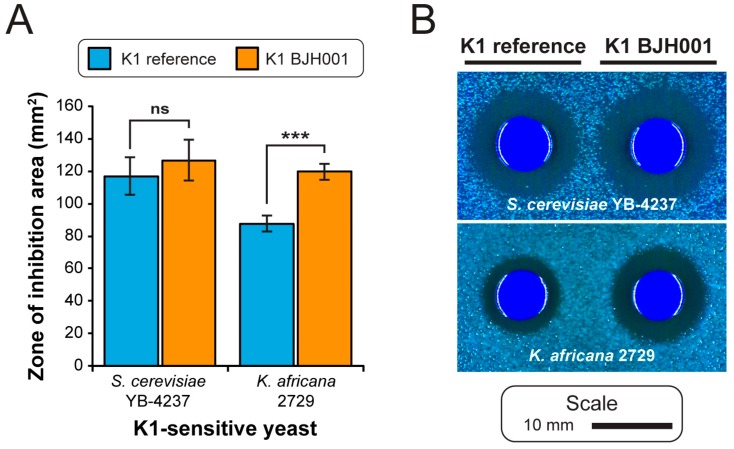
Mutations within the K1 gene increases the ability of the killer toxin to inhibit the growth of the yeast *Kazachstania africana in vitro*. (**A**) The change in the area of growth inhibition around K1-expressing *S. cerevisiae* challenged with different strains of K1-sensitive yeasts measured in mm^2^. Asterisks are indicative of a significant difference in the mean zone of the inhibition area (T-test, two-tailed, *** *p* < 0.01, ns indicates no significant difference). Error bars represent standard error of three independent repeats. (**B**) Representative images of the isogenic non-killer yeast strains expressing different K1 killer toxins (derived from the K1 reference sequence or K1 from *S. cerevisiae* BJH001), on agar seeded with yeasts known to be sensitive to K1 killer toxins.

**Table 1 viruses-11-00070-t001:** Sequencing and assembly statistics for libraries made from dsRNAs polyadenylated with varying poly(A) polymerase concentrations.

Criteria	25 Units Poly(A) Polymerase	0.02 Units Poly(A) Polymerase	Fold Increase
Number of reads	22,521	471,742	21
Homopolymeric reads	68%	0.6%	0.01
De novo assembled contigs > 100nt	21	98	5
Length of largest contig	925	5022	5
Mean coverage of viral contigs	1	610	610

**Table 2 viruses-11-00070-t002:** Polymorphic sites in endogenous dsRNA virus genomes from a clonal isolate of the *S. cerevisiae* strain BJH001.

dsRNA Element	Start Position (bp)	Polymorphism Type	Mutation	Coding Change	Coverage	Freq. (%)
ScV-L-A1	89	SNP (transition)	U to C	Non-synonymous	1002	5.8
ScV-L-A1	269	Indel	Insertion U	Gag truncation	1332	5.4
ScV-L-A1	407	SNP (transition)	C to U	Synonymous	848	31.8
ScV-L-A1	1526	SNP (transition)	U to C	Synonymous	1297	7.9
ScV-L-A1	2362	Indel	Insertion A	+1 frameshift	583	20.8
ScV-L-A1	2417	Indel	Deletion U	-1 frameshift	618	20.9
ScV-L-A1	2860	SNP (transition)	G to A	Synonymous	43,300	68.4
ScV-L-A1	4557	Indel	Insertion A	n/a	140	7.1
ScV-L-BC	10	SNP (transition)	G to A	n/a	245	81.6
ScV-L-BC	10	SNP (transversion)	G to U	n/a	245	8.6
ScV-L-BC	455	SNP (transition)	G to A	Synonymous	1882	8.3
ScV-L-BC	884	SNP (transition)	A to G	Synonymous	13,398	8.9
ScV-L-BC	1551	Indel	Insertion U	Gag truncation	207	8.9
ScV-L-BC	1569	Indel	Deletion U	Gag truncation	203	13.3
ScV-L-BC	3424	Indel	Insertion U	Pol truncation	8674	5.4

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
