# Peer review of "A Rapid Method for Sequencing Double-Stranded RNAs Purified from Yeasts and the Identification of a Potent K1 Killer Toxin Isolated from Saccharomyces cerevisiae"

_viruses, 2019, doi:10.3390/v11010070_

Reviewer 1 Report

Review: Viruses-390803

In this manuscript, the Crabtree et al. have improved the methodology for the sequencing dsRNA viruses from Saccharomyces.  The authors demonstrate their protocol with two viruses carried by a strain of S. cerevisiae.  They are able to detect mutations within the viral population, including mutations within a cytotoxin gene that modulates toxicity against a specific yeast species.  This work extends current techniques and will be of interest to researchers studying mycoviruses.

Addressing the following comments will improve the manuscript:

Line 94: It is unclear how yeast cells were lysed. Please provide a more detailed description. Were frozen cells were disrupted by mortar and pestle as in Okada et al, or by another method?

Line 154: Details about the fluorometer protocol should be included- is this the plate reader in line 141, were there any modifications to the manufacturers protocol?

Line 201: “To quantitatively compare…” is not a complete sentence.

Line 211: Fix capitalization of “dsRNA”

Figure 1A: The seeding of lawn used in killer assays appears very uneven and the image color is inconsistent (e.g. Figure 1A, more pronounced in Figure S3)

Figure 1A: The authors should consider stating in the text, legend, or a table the expected killer phenotype of each yeast used.

Figure 1B: The legend needs to indicate that the gel shows RNA purified with the new extraction protocol.

Figure 4: This figure is very difficult to interpret.  We recommend the authors include a visual legend illustrating what each color and shape indicates.  Furthermore, the authors should use principles of color universal design (jfly.iam.u-tokyo.ac.jp/color) and choose a color-blind friendly palette (e.g. www.nature.com/articles/nmeth.1618).

Table 2:  A table containing the complete list of fixed and polymorphic mutations should be included as a supplemental file.

Line 322: The authors identify indel pairs within a single virus, indicating they have reads spanning both mutations which allows the differentiation of co-occurring mutations within a single virus as opposed to two single mutations in the viral population occurring in different viruses.  This should be clarified in the text.

Line 388:  Need to insert reference.

Line 397:  This sentence is unclear as written. Should it read “reference sequence-independent”?

Line 406:  The authors do not previously state they confirmed fixed mutations in the K1 gene by Sanger sequencing in the Results section.  This should be included in the Results or Supplement with appropriate methods included.

 Author Response

RESPONSE TO REVIEWER 1

In this manuscript, the Crabtree et al. have improved the methodology for the sequencing dsRNA viruses from Saccharomyces.  The authors demonstrate their protocol with two viruses carried by a strain of S. cerevisiae.  They are able to detect mutations within the viral population, including mutations within a cytotoxin gene that modulates toxicity against a specific yeast species.  This work extends current techniques and will be of interest to researchers studying mycoviruses.

Addressing the following comments will improve the manuscript:

Line 94: It is unclear how yeast cells were lysed. Please provide a more detailed description. Were frozen cells were disrupted by mortar and pestle as in Okada et al, or by another method?

Done  - Sentence clarified: “To extract dsRNAs, 450 μL of 2x LTE [500 mM LiCl; 20 mM Tris-HCl, pH 8.0; 30 mM EDTA, pH 8.0] containing 0.1 % (v/v) beta-mercaptoethanol (14.3 M) (Amresco) was added to harvested yeast cells. The cell mixture was vortexed for 3 minutes at 3000 rpm (Disruptor Genie, Scientific Industries). 50 μL of 10% (w/v) SDS solution and 500 μL of phenol–chloroform–isoamyl alcohol [25:24:1] pH 8.0 were added to the crude cell extracts and vortexed until homogenous.”

Line 154: Details about the fluorometer protocol should be included- is this the plate reader in line 141, were there any modifications to the manufacturers protocol?

Done – additional details on the method included in section 2.3: “All cDNA samples were normalized to 2.5 ng/μL for the desired final average library insert size of 550 bp. Fluorometric quantification was performed with SpectraMax Gemini XPS plate reader (Molecular Devices) and PicoGreen (Invitrogen). For fluorometric quantification, 2 μL of cDNA was diluted in 98 μL 1x TE buffer (10 mM Tris-HCl, 1 mM EDTA, pH 7.5), and mixed with 100 μL of PicoGreen (diluted 1:200 in TE). Standards were prepared per manufacturers protocol and scaling volumes to 1/10th. Samples and standards were incubated at ambient temperature in the dark for 5 minutes before analysis.”

Line 201: “To quantitatively compare…” is not a complete sentence.

Done - Sentence rewritten: “To quantitatively compare the antifungal activities of the different K1 toxins, single colonies of S. cerevisiae transformed with the plasmids pEK005 and pEK006 were inoculated in 1 mL of complete liquid media lacking uracil with 2% galactose. These cultures were incubated at ambient temperature for 48 hours with vigorous shaking. K1 toxin-susceptible yeasts were inoculated in 1 mL of YPD and incubated at ambient temperature for 48 hours with vigorous shaking..”

Line 211: Fix capitalization of “dsRNA”

            Done

Figure 1A: The seeding of lawn used in killer assays appears very uneven and the image color is inconsistent (e.g. Figure 1A, more pronounced in Figure S3)

            Done - Figure 1A revised to better display the pictures of the killer yeast.

Figure 1A: The authors should consider stating in the text, legend, or a table the expected killer phenotype of each yeast used.

Done - text changed to include a description of known and unknown killer yeasts: “The presence of dsRNA mycoviruses in S. cerevisiae is often correlated with the production of antifungal proteins (killer toxins) [30]. We used two yeast strains that have been previously reported as killer yeasts (S. cerevisiae BJH001 [33] and S. paradoxus Y8.5 [34,35]), one non-killer yeast (S. paradoxus CBS432) and several commercially-available dried yeasts to assay the effectiveness of a modified protocol based on a dsRNA extraction method previously optimized for filamentous fungi and plant material (Figure 1) [28].

Figure 1B: The legend needs to indicate that the gel shows RNA purified with the new extraction protocol.

Done – text added: “(b) dsRNAs purified by the method described in this paper were separated by 0.8% agarose gel electrophoresis and stained with ethidium bromide.”

Figure 4: This figure is very difficult to interpret.  We recommend the authors include a visual legend illustrating what each color and shape indicates.  Furthermore, the authors should use principles of color universal design (jfly.iam.u-tokyo.ac.jp/color) and choose a color-blind friendly palette (e.g. www.nature.com/articles/nmeth.1618).

Done – Figure has been reorganized, a visual legend added to the figure and colors changed to be more mindful of people with different perceptions of color. Text legend altered to remove redundancy with visual legend.

Table 2:  A table containing the complete list of fixed and polymorphic mutations should be included as a supplemental file.

Done – We now include this information as a supplementary file S2.

Line 322: The authors identify indel pairs within a single virus, indicating they have reads spanning both mutations which allows the differentiation of co-occurring mutations within a single virus as opposed to two single mutations in the viral population occurring in different viruses.  This should be clarified in the text.

Done – additional text added: “Two polymorphic indels that are present together in 21% of ScV-L-A1 dsRNAs cause a +1 frameshift followed after 55 base pairs by a -1 frameshift. These nucleotide changes always occur together on the same sequence read, meaning that the first frameshift mutation (the insertion of an adenine nucleotide) is corrected by the second mutation (the deletion of a uracil nucleotide) and the reading frame is maintained. In the context of ScV-L-A1, this results in an 18 amino acid frameshift in the Gag-Pol fusion protein (Figure 4A). Similarly, two indels (one nucleotide insertion and one deletion) in ScV-L-BC result in a small fixed 4 amino acid frameshift within the C-terminus of the Gag-Pol fusion protein (Figure 4B).”

Line 388:  Need to insert reference.

            Done – incorrectly positioned placeholder removed

Line 397:  This sentence is unclear as written. Should it read “reference sequence-independent”?

Done – section rewritten: “The low complexity of this sequence may have resulted in ambiguities during sequence mapping that could have caused poor sequence coverage. Except for the low coverage of the terminal nucleotides, we were able to assemble long viral contigs from mixtures of different dsRNAs extracted from a single strain of yeast, independent of a reference sequence (Figure 3B). For example, we have previously studied the mycoviruses and dsRNAs present within S. cerevisiae BJH001 by agarose gel electrophoresis but were unaware of the presence of a variant ScV-L-BC within this strain because of its similar electrophoretic mobility to ScV-L-A1 [33].”

Line 406:  The authors do not previously state they confirmed fixed mutations in the K1 gene by Sanger sequencing in the Results section.  This should be included in the Results or Supplement with appropriate methods included.

Done – text added to section 3.3: “Importantly, the four mutations identified within the BJH001 K1 gene by Illumina NGS were confirmed by Sanger sequencing.”

Reviewer 2 Report

In the present study, Crabtree et al. present method for sequencing of dsRNA found in S. cerevisiae BJH001 and analysis of found viruses. The manuscript describes procedures of preparation of source genetic material from the yeast cells, preparation of cDNA for sequencing and addresses the sequence variation within virus population with following functional evaluation of mutation set found in M1 dsRNA from strain BJH001.

Discovery of new dsRNA-based mycoviruses, in particularly those from yeast, attracts much attention nowadays. The availability of NGS allows robust and deep analysis of virtually any genetic material, once required protocols are established. This study aims on the deep analysis of several dsRNAs found within individual yeast strain, a promising development for understanding of various aspects of virus maintenance.

 Minor remarks and observations are as follow:

 Lanes 10-11 and 16-17: punctuation (use of semicolons) unjustified.

Lanes 12, 14, 16: style should be improved by avoiding unnecessary repetitions of “many”.

Lane 29-30: the last statement of the Abstract has poor if any connection with the research described in the manuscript.

Lane 71: dilution of enzyme is barely an enzyme optimization per se.

Lane 93: pipetting of 0.45 mkl volumes is confusing. Exact concentration of 2-mercaptoethanol in the final mixture would be more descriptive.

Lane 116: double space present.

Lane 120-121: more detailed description on polyA addition is necessary.

Lanes 224-226: references should be provided in corresponding positions of the sentence to avoid confusion interpreting them.

Lane 227: yeast strains used for dsRNA extraction should be described in Materials section.

Lane 230: figure number in picture is redundant (here and the rest Figures), „Dried yeast“ is  underscored - why?

Lane 388: reference is missing.

Lane 392-393: how was 5‘ termini of ScV-M1 and ScV-K1 polyadenylated?

Lane 416: “...studies...“ is a typo.

 In addition, there are several major observations to be considered before publishing this material.

 1. The same killer toxin, synthesized either from dsRNR or a plasmid (Lane 339-341), has not been demonstrated to change or alter its specificity toward sensitive yeast species so far. Therefore, the present statement should be reformulated to better reflect the idea of this sentence. Also, expression of killer from the plasmid (Lane 337-339) might affect it’s amount - and therefore the overall efficiency of toxin action, not the specificity as stated. The amount of information supplied for plasmid constructs precludes the fine resolution between these possibilities.

2. L-BC identification (Lane 398-341) is merely a big discovery nowadays, given the genome sequences present in Genbank (items NC_001641, KX906605, KT784813); therefore, much faster and easier observation would be by RT-PCR. It’s fine to cite it as by-product of NGS experiment, however other dsRNAs – that of L-A-1 and M1 were known for some time, too.

3. Combination of methods engaged in this study (Lane 384-386) might be so far unreported indeed, however it is the combination of already described techniques, none of which have been described for the first time in this manuscript. Also, polyadenylation of 3’ terminus has been described for viral RNA sequencing, as properly stated in a manuscript and elsewhere (Grybchuk et al, 2018). Even if the NGS previously has not been invoked for sequencing, the proof-of-principle is there for some time already and therefore not novel.

4. The involvement of Illumina’s NGS, even relatively low-throughput MiSeq, might as well hinder the true composition of viriome in a cell. In particular, this observation is based on presence of polymorphic indels, located just below the frameshift region, in 23% of Sc-L-A1 virus sequences determined. In combination of extremely high coverage, reaching median read depth of more than 2500, this requires to address the reason behind such high variability of virus, known to play an essential role in killer phenotype maintenance. Is it the true intracellular variability, or merely the reflection of shortage of reverse transcriptase to overcome secondary structures of the frameshift region during cDNA synthesis? Unfortunately, this point remains un-addressed in the manuscript. NGS approach might therefore multiply the false mutations progressively. In addition, coverage at thousands fold for few different dsRNA molecules might appear to be the unnecessary waste of resources, too.

In summary, this manuscript includes combination of published methods for sequencing, resulting in complete workflow for sequence determination of dsRNA from yeast. Hardly there is a sound scientific novelty, while scientific community might still benefit. Reformulated focus of this study should make it more attractive for publishing.

Author Response

RESPONSE TO REVIEWER 2

In the present study, Crabtree et al. present method for sequencing of dsRNA found in S. cerevisiae BJH001 and analysis of found viruses. The manuscript describes procedures of preparation of source genetic material from the yeast cells, preparation of cDNA for sequencing and addresses the sequence variation within virus population with following functional evaluation of mutation set found in M1 dsRNA from strain BJH001.

Discovery of new dsRNA-based mycoviruses, in particularly those from yeast, attracts much attention nowadays. The availability of NGS allows robust and deep analysis of virtually any genetic material, once required protocols are established. This study aims on the deep analysis of several dsRNAs found within individual yeast strain, a promising development for understanding of various aspects of virus maintenance.

 Minor remarks and observations are as follow:

 Lanes 10-11 and 16-17: punctuation (use of semicolons) unjustified.

The original submitted document did not contain these semicolons. We assume that they were added during reformatting to mdpi style.
Done – punctuation altered to read:
Mycoviruses infect a large number of diverse fungal species but considering their prevalence, relatively few high-quality viral genome sequences have been determined.

Lanes 12, 14, 16: style should be improved by avoiding unnecessary repetitions of “many”.

Done – section rewritten: “Many mycoviruses have linear double-stranded RNA genomes that makes it technically challenging to ascertain their nucleotide sequence using conventional sequencing methods. Different specialist methodologies have been developed for the extraction of double-stranded RNAs from fungi and the subsequent synthesis of cDNAs for cloning and sequencing. However, these methods are often labor intensive, time consuming, and often requiring several days to produce cDNAs from double-stranded RNAs.”

Lane 29-30: the last statement of the Abstract has poor if any connection with the research described in the manuscript.

Done – sentence deleted

Lane 71: dilution of enzyme is barely an enzyme optimization per se.

Done – We show in table 1 and in the accompanying text the critical importance of optimizing conditions for 3’ end tailing because of its impact on NGS run quality and data analysis. This procedure of determining the optimal concentration of poly(A) polymerase required several NGS runs and many personnel hours, which demonstrates that this optimization was not a trivial undertaking. We have expanded this sentence to describe the additional optimization of our reported method. “However, there remain opportunities to improve the efficiency of dsRNA sequencing techniques by optimizing dsRNA purification, reverse transcription, and 3’ end tailing to improve the quality of cDNAs used for NGS sequencing.”

Lane 93: pipetting of 0.45 mkl volumes is confusing. Exact concentration of 2-mercaptoethanol in the final mixture would be more descriptive.

Done – text edited to clarify the protocol. “To extract dsRNAs, 450 μL of 2x LTE [500 mM LiCl; 20 mM Tris-HCl, pH 8.0; 30 mM EDTA, pH 8.0] containing 0.1 % (v/v) beta-mercaptoethanol (14.3 M) (Amresco) was added to harvested yeast cells.”

Lane 116: double space present.

Done – all double spaces removed from the manuscript

Lane 120-121: more detailed description on polyA addition is necessary.

Done – additional details of the method added: “Poly(A) polymerase (New England Biolabs) was used to synthesize a poly(A) tail at the 3’ termini of all denatured dsRNAs. To 12.5 μL of purified dsRNAs, the following was added: 1.5 μL 10x poly(A) polymerase reaction buffer, 1.5 μL adenosine 5’ triphosphate [10 mM], 0.5 μL of poly(A) polymerase (diluted 1:32 in nuclease-free water), and 0.5 μL murine RNAse inhibitor. Samples were incubated at 37°C for 30 minutes, 65°C for 20 minutes, 98°C for 5 minutes, and then immediately placed in a wet ice slurry.”

Lanes 224-226: references should be provided in corresponding positions of the sentence to avoid confusion interpreting them.

Response – both of these articles are reviews that describe mycoviruses in yeasts and killer toxins.

Lane 227: yeast strains used for dsRNA extraction should be described in Materials section.

Done – we now include a supplementary table S2 that lists all strains of yeast used in this study .

Lane 230: figure number in picture is redundant (here and the rest Figures), „Dried yeast“ is  underscored - why?

            Done – All figures updated

Lane 388: reference is missing.

Done – incorrectly positioned placeholder removed

Lane 392-393: how was 5‘ termini of ScV-M1 and ScV-K1 polyadenylated?

Response – the 5’ termini would not be polyadenylated because of the known specificity of the enzyme for 3’ termini.  There is no enzymatic tailing of the 5’ termini in this paper. The addition of the poly(T) tracts to the 5’ termini (diagrammed in Figure 2A) is the result of repair of cDNAs by polymerases and subsequent amplification by PCR.

Lane 416: “...studies...“ is a typo.

            Done – typo corrected

In addition, there are several major observations to be considered before publishing this material.

 1. The same killer toxin, synthesized either from dsRNR or a plasmid (Lane 339-341), has not been demonstrated to change or alter its specificity toward sensitive yeast species so far. Therefore, the present statement should be reformulated to better reflect the idea of this sentence. Also, expression of killer from the plasmid (Lane 337-339) might affect it’s amount - and therefore the overall efficiency of toxin action, not the specificity as stated. The amount of information supplied for plasmid constructs precludes the fine resolution between these possibilities.

Done – We have worked on clarifying this section of the manuscript to reflect that plasmid vs dsRNA expression does not alter the specificity of K1 killer toxins, but the mutations identified within BJH001 K1 can improve its antifungal activity against K. africanus: “To compare the biological activities of the two cloned K1 toxins, 6 x 106 cells of each isogenic K1-expressing strain were spotted in triplicate onto galactose-containing agar plates seeded with various K1-sensitive yeasts. Qualitative comparison of the specificity of BJH001 K1 toxin expressed from a plasmid or dsRNA satellite demonstrated that ectopic expression does not alter its specificity toward toxin-sensitive yeast (Figure S3 and S4). However, measurement of the area of growth inhibition revealed that the BJH001 K1 toxin produces significantly larger zones of growth inhibition on the K1-sensitive yeast Kazachstania africana compared to the canonical K1 reference toxin (T-test, two-tailed, p<0.01) (Figure 5 and S4). This zone of growth inhibition is 28% larger than the K1 reference toxin (Figure 5). For the seven other K1-sensitive strains tested, the differences between the two killer toxins did not significantly alter the area of the zone of growth inhibition, which suggests that the mutations in the K1 from strains BJH001 do not affect the amount of killer toxin produced or the rate of diffusion through the agar. These data demonstrate that mutations in K1 killer toxins can alter their specificity for different species of yeasts.”

2. L-BC identification (Lane 398-341) is merely a big discovery nowadays, given the genome sequences present in Genbank (items NC_001641, KX906605, KT784813); therefore, much faster and easier observation would be by RT-PCR. It’s fine to cite it as by-product of NGS experiment, however other dsRNAs – that of L-A-1 and M1 were known for some time, too.

Done – we have edited this section to better communicate the fact that our method was able to detect and fully sequence a virus that we did not expect to find within BJH001. We were not surprised by the presence of L-BC, it was that we did not realize that it was in a strain that we have been working with for many years! “Except for the low coverage of the terminal nucleotides, we were able to assemble long viral contigs from mixtures of different dsRNAs extracted from a single strain of yeast, independent of a reference sequence (Figure 3B). For example, we have previously studied the mycoviruses and dsRNAs present within S. cerevisiae BJH001 by agarose gel electrophoresis but were unaware of the presence of a variant ScV-L-BC within this strain because of its similar electrophoretic mobility to ScV-L-A1 [33]. By applying our NGS method we were able to identify the presence of ScV-L-BC and assemble a large contig with high similarity to the reference sequence of this totivirus (Figure 3).”

3. Combination of methods engaged in this study (Lane 384-386) might be so far unreported indeed, however it is the combination of already described techniques, none of which have been described for the first time in this manuscript. Also, polyadenylation of 3’ terminus has been described for viral RNA sequencing, as properly stated in a manuscript and elsewhere (Grybchuk et al, 2018). Even if the NGS previously has not been invoked for sequencing, the proof-of-principle is there for some time already and therefore not novel.

Response – We understand the reviewer’s point of view, but we would argue that the leveraging of Illumina NGS methods to successfully sequence polyadenylated dsRNAs is an efficient method that needs to be reported to the wider scientific community (which the reviewer agrees with in their summary statement below). We outline the methods in great detail so that any researcher should be able to benefit from our optimization of the technique. We demonstrate in the manuscript that it is technically challenging to determine the optimal conditions to combine 3’ polyadenylation and NGS.

4. The involvement of Illumina’s NGS, even relatively low-throughput MiSeq, might as well hinder the true composition of viriome in a cell. In particular, this observation is based on presence of polymorphic indels, located just below the frameshift region, in 23% of Sc-L-A1 virus sequences determined. In combination of extremely high coverage, reaching median read depth of more than 2500, this requires to address the reason behind such high variability of virus, known to play an essential role in killer phenotype maintenance. Is it the true intracellular variability, or merely the reflection of shortage of reverse transcriptase to overcome secondary structures of the frameshift region during cDNA synthesis? Unfortunately, this point remains un-addressed in the manuscript. NGS approach might therefore multiply the false mutations progressively. In addition, coverage at thousands fold for few different dsRNA molecules might appear to be the unnecessary waste of resources, too.

Done – In response to the reviewer’s comments that we should be further exploring these interesting mutations, we feel that this goes beyond the scope of this study which is primarily focused on reporting a new sequencing methodology and the description of a potent variant of the K1 killer toxin. However, we have added a new section in the discussion to better describe our observations of the polymorphic indels: “We observed several indels that result in small, but dramatic changes to the amino acid sequence of the polymerase domain in both ScV-L-A1 and ScV-L-BC. These frameshifted regions are peripheral to the conserved motifs of the catalytic core of the RNA-dependent RNA polymerase [40], which would suggest that these mutations may not disrupt polymerase function. Furthermore, because ScV-L-BC is stably associated with S. cerevisiae BJH001, we do not expect that the fixed 4 amino acid frameshift to significantly affect virus persistence. The 18 amino acid frameshift identified within ScV-L-A1 is polymorphic, thus it is harder to predict whether these indels cause defects in polymerase activity. Frameshift mutations most often result in premature stop codons and defective proteins that are truncated, with the most prominent examples of frameshift mutations being those that cause human disease [41-43]. Moreover, mutant polymerases could be incorporated into viral capsids to form defective interfering particles with likely negative consequences for viral replication. Alternatively, frameshift mutations can result in novel protein functions, although they are less frequently reported [44]. The co-occurrence of +1 and -1 frameshift indels suggests that there has been selection to maintain the reading frame of the polymerase gene, but any functional consequence to the enzyme and on the replication and persistence of ScV-L-A1 remains unexplored.”

It is extremely unlikely that errors in the replication of the dsRNA would result in concurrent mutations occurring in 20% of all sequenced RT reads. Moreover, the 3' tailing that we describe in our method would allow the initiation of reverse transcription on both strands of the dsRNA which would prevent any sequencing bias in this polymorphic region due to the presence of secondary structure in the frameshift region.

In summary, this manuscript includes combination of published methods for sequencing, resulting in complete workflow for sequence determination of dsRNA from yeast. Hardly there is a sound scientific novelty, while scientific community might still benefit. Reformulated focus of this study should make it more attractive for publishing.

 Reviewer 3 Report

This manuscript is dealing with an interesting topic which is the difficulty to sequence dsRNA viral genomes. The authors claim that they have developed a rapid method for the purification and sequencing of dsRNA mycoviruses that optimizes the extraction of high-quality double stranded RNAs from yeasts and 3’ polyadenylation for the initiation of cDNA synthesis for next generation sequencing. Additionally, they have found a new version of M1 killer virus that shows more intense killer activity than previously known reference K1 virus. This last interesting issue is not mentioned in the title of the manuscript, just a related sentence is in the abstract, it is poor discussed in the discussion section, but many related results are shown. My major concern is that the authors do not precisely compare their method with similar methods already published, some of them are not even referred in this manuscript (some examples being: Ramírez, M.; Velázquez, R.; Maqueda, M.; López-Piñeiro, A.; Ribas, J.C. A new wine Torulaspora delbrueckii killer strain with broad antifungal activity and its toxin-encoding double-stranded RNA virus. Front. Microbiol. 2015, 6, 983., Ramirez, M.; Velazquez, R.; Lopez-Pineiro, A.; Naranjo, B.; Roig, F.; Llorens, C. New insights into the genome organization of yeast killer viruses based on "atypical" killer strains characterized by high-throughput sequencing. Toxins (Basel) 2017, 9.). This point should be addressed before this manuscript is acceptable for publication. Here are some suggestions that could be considered for manuscript improvement:

 Title

-Consider to mention the finding of new mutations in K1 genome that increase killer phenotype intensity.

 Introduction

-Lines 69-81. This paragraph is related to results, it is not aims and scope. Move to elsewhere. Maybe part of the paragraph can be moved to the abstract and part of it to the discussion/conclusions section.

 Materials and Methods

-Information related to yeast strains used in this work is missing.

 Results

-Line 223. “….and phenol that we previously described (Figure S1)”. The authors should show the full length of the gel to evaluate the mtDNA (75-80 kpb) contamination of the sample.

 -Lines 231-237. Fig. 1 legend. Which is the yeast strain used as lawn in this killer test?

 -Line 303. Figure 3. Central Poly(A) region of M1 genome was not covered using this sequencing strategy? Mention and discus. Compare with other similar strategies already published leading with next-generation sequencing of dsRNA M viruses from S. cerevisiae.

 -Line 322. Move “(Table 2)” to the precedent sentence since no data related to M1 “mutations” is given in this table, or even better, include these missing data in Table 2.

 -Line near 325. The authors should give the sequences of virus genomes, indicating the position of each nucleotide change with respect to the reference viruses, in supplementary materials, and refer each mutation somewhere in this paragraph. This will facilitate the precise location of each mutation by the readers.

 -Lines 361-363. Data for M1 virus are missing in Table 2. Include.

 -Lines 365-371. Which is the K1 reference strain? Mention it.

 Discussion

-Lines 387-389. The authors should double check this affirmation. Again, see references: Ramírez, M.; Velázquez, R.; Maqueda, M.; López-Piñeiro, A.; Ribas, J.C. A new wine Torulaspora delbrueckii killer strain with broad antifungal activity and its toxin-encoding double-stranded RNA virus. Front. Microbiol. 2015, 6, 983., Ramirez, M.; Velazquez, R.; Lopez-Pineiro, A.; Naranjo, B.; Roig, F.; Llorens, C. New insights into the genome organization of yeast killer viruses based on "atypical" killer strains characterized by high-throughput sequencing. Toxins (Basel) 2017, 9.

 -Lines 401-403. Similar results have been reported before, see references above. Mention and discuss.

 -Lines 409-411. It is hard to me seeing the cytotoxicity increase of K1 toxin in Fig. S4. The figure is very well done but these results are not relevant. Consider to remove this paragraph and Fig. S4.

 -Lines 406-412. This issue, that seems very interesting, is poorly discussed in the manuscript.

Author Response

RESPONSE TO REVIEWER 3

 This manuscript is dealing with an interesting topic which is the difficulty to sequence dsRNA viral genomes. The authors claim that they have developed a rapid method for the purification and sequencing of dsRNA mycoviruses that optimizes the extraction of high-quality double stranded RNAs from yeasts and 3’ polyadenylation for the initiation of cDNA synthesis for next generation sequencing. Additionally, they have found a new version of M1 killer virus that shows more intense killer activity than previously known reference K1 virus. This last interesting issue is not mentioned in the title of the manuscript, just a related sentence is in the abstract, it is poor discussed in the discussion section, but many related results are shown. My major concern is that the authors do not precisely compare their method with similar methods already published, some of them are not even referred in this manuscript (some examples being: Ramírez, M.; Velázquez, R.; Maqueda, M.; López-Piñeiro, A.; Ribas, J.C. A new wine Torulaspora delbrueckii killer strain with broad antifungal activity and its toxin-encoding double-stranded RNA virus. Front. Microbiol. 2015, 6, 983., Ramirez, M.; Velazquez, R.; Lopez-Pineiro, A.; Naranjo, B.; Roig, F.; Llorens, C. New insights into the genome organization of yeast killer viruses based on "atypical" killer strains characterized by high-throughput sequencing. Toxins (Basel) 2017, 9.). This point should be addressed before this manuscript is acceptable for publication. Here are some suggestions that could be considered for manuscript improvement:

 Title

-Consider to mention the finding of new mutations in K1 genome that increase killer phenotype intensity.

Done – Title altered to: “A Rapid Method for Sequencing dsRNAs Purified from Yeasts and the Identification of a Potent K1 Killer Toxin Isolated from S. cerevisiae.

Introduction

-Lines 69-81. This paragraph is related to results, it is not aims and scope. Move to elsewhere. Maybe part of the paragraph can be moved to the abstract and part of it to the discussion/conclusions section.

Done – this section has been edited: “Here we report a comprehensive approach that leverages NGS technologies to determine the genetic sequence of dsRNAs purified from yeasts. Specifically, we combine rapid dsRNA extraction and cDNA synthesis protocols to create high quality cDNAs for downstream NGS library preparation using an affordable ‘tagmentation’ procedure. We demonstrate the optimization of poly(A) polymerase for the 3' polyadenylation of viral dsRNAs for ‘anchored’ oligo(dT) priming, which simplifies the reverse-transcription, amplification, and sequencing of cDNAs by NGS. This method has allowed the detailed description of the diversity of dsRNAs found within a single strains of S. cerevisiae.”

Materials and Methods

-Information related to yeast strains used in this work is missing.

Done – we have added a new supplementary table S2 that details all of the yeast strains used in this study and their source.

Results

-Line 223. “….and phenol that we previously described (Figure S1)”. The authors should show the full length of the gel to evaluate the mtDNA (75-80 kpb) contamination of the sample.

Done – we now include all of the raw gel images in the manuscript in the supplementary material (File S3). In both cases DNA contamination was slight or not detectable and subsequently removed by DNase treatment prior to 3’ tailing and cDNA synthesis.

-Lines 231-237. Fig. 1 legend. Which is the yeast strain used as lawn in this killer test?

Done – we have included details of the lawn strain used in Figure 1 (S. bayanus CBS7001).

-Line 303. Figure 3. Central Poly(A) region of M1 genome was not covered using this sequencing strategy? Mention and discus. Compare with other similar strategies already published leading with next-generation sequencing of dsRNA M viruses from S. cerevisiae.

Response – we include a sentence describing the reason for the low coverage in this region in the results section: “The only other region that was not resolved was the low complexity ~200 bp homopolymeric adenosine-rich tract contained within the ScV-M1 satellite dsRNA, which was masked prior to read mapping.”

-Line 322. Move “(Table 2)” to the precedent sentence since no data related to M1 “mutations” is given in this table, or even better, include these missing data in Table 2.

            Done – “(table 2)” moved to a more appropriate position in the text

-Line near 325. The authors should give the sequences of virus genomes, indicating the position of each nucleotide change with respect to the reference viruses, in supplementary materials, and refer each mutation somewhere in this paragraph. This will facilitate the precise location of each mutation by the readers.

Done – we now include an additional data file that details all of the mutations observed in our study within each virus and satellite (File S2).

-Lines 361-363. Data for M1 virus are missing in Table 2. Include.

Response – table 2 is showing polymorphic mutations.  None were detected within the populations of M1 dsRNAs within BJH001, as shown in Figure 4 and written the in the text: “No polymorphic nucleotides were found in assembled contigs for the ScV-M1 dsRNA.”

-Lines 365-371. Which is the K1 reference strain? Mention it.

Done – we now include the GenBank accession numbers earlier in the manuscript: Of the 471,742 reads sequenced for this sample, 97.84% could be mapped against viral sequences (NCBI GenBank accession numbers: ScV-L-A-1, M28353.1; ScV-L-BC, NC_001641.1; ScV-M1, NC_001782.1).”

Discussion

-Lines 387-389. The authors should double check this affirmation. Again, see references: Ramírez, M.; Velázquez, R.; Maqueda, M.; López-Piñeiro, A.; Ribas, J.C. A new wine Torulaspora delbrueckii killer strain with broad antifungal activity and its toxin-encoding double-stranded RNA virus. Front. Microbiol. 2015, 6, 983., Ramirez, M.; Velazquez, R.; Lopez-Pineiro, A.; Naranjo, B.; Roig, F.; Llorens, C. New insights into the genome organization of yeast killer viruses based on "atypical" killer strains characterized by high-throughput sequencing. Toxins (Basel) 2017, 9.

Done – We appreciate the reviewers providing these additional references, we now add them (and additional references) to the manuscript: “The 3’ polyadenylation of cellular mRNAs can be used to prime cDNA synthesis using oligo(dT) primers, but the 3’ termini of linear mycoviral dsRNAs lack these terminal homopolymeric sequences. However, homopolymeric adenosine-rich tracts that are known to be encoded internally by satellite dsRNAs found within yeasts have been successfully used to prime cDNA synthesis [12-14].” and “Cloning of dsRNA-derived cDNAs and 5’or 3’ RACE enable the efficient resolution of dsRNA terminal ends. NGS methods alone have been largely unsuccessful in the resolution of the terminal ends of cDNAs [21,38]. However, there are some notable exceptions that have leveraged a combination of commercial kits and homopolymeric primers to completely sequence dsRNAs isolated from yeasts [12,13].”

-Lines 401-403. Similar results have been reported before, see references above. Mention and discuss.

Done – based on comments from several reviewers, we have rewritten to clarify the intended meaning of this section to better communicate the use of our method to identify multiple dsRNAs in a single sample: “This will enable the application of this sequence-independent method to identify novel mycoviruses and satellite dsRNAs found within fungi. For example, we have previously studied the mycoviruses and dsRNAs present within S. cerevisiae BJH001 by agarose gel electrophoresis but were unaware of the presence of a variant ScV-L-BC within this strain because of its similar electrophoretic mobility to ScV-L-A1. By applying our NGS method we were able to identify the presence of ScV-L-BC and assemble a large contig with high similarity to the reference sequence of this totivirus (Figure 3).”

-Lines 409-411. It is hard to me seeing the cytotoxicity increase of K1 toxin in Fig. S4. The figure is very well done but these results are not relevant. Consider to remove this paragraph and Fig. S4.

Done – we have reorganized this section to better communicate these negative data and their relevance in Figure S4 and we have also adjusted the contrast of the figure to improve clarity: “For the seven other K1-sensitive strains tested, the differences between the two killer toxins did not significantly alter the area of the zone of growth inhibition, which suggests that the mutations in the K1 toxin from strain BJH001 does not affect the amount of killer toxin produced or the rate of diffusion through the agar (Figure S4). These data demonstrate that mutations in K1 killer toxin can alter their specificity for different species of yeasts.”

-Lines 406-412. This issue, that seems very interesting, is poorly discussed in the manuscript. Improve.

Done – we have expanded the discussion of these results: “The two non-synonymous mutations (I103S and T146I) map to the K1 alpha-domain that is important for the cytotoxicity of K1 and are positioned close to known mutations that are defective in target cell wall binding and toxicity (D101R and D140R) (Figure 4D). Relative to the cloned canonical K1 toxin, these mutations significantly increased the toxicity of K1 to K. africana but not to other strain of Saccharomyces yeasts that were challenged by the toxin (Figure 5). Previously, two different K1 variants have been described within species of Saccharomyces yeasts by reverse transcriptase PCR and Sanger sequencing. Expression of these variant K1 toxins in S. cerevisiae and S. paradoxus appeared to show that a single gain-of-function mutation in the K1 beta-domain (L251F) can increase the cytotoxicity of the K1 toxin, but the results were not quantified to assess statistical significance. Furthermore, a separate study failed to identify these K1 variants in the same strains of yeast. We anticipate that our NGS method could be applied to rapidly elucidate the genetic sequence of satellite dsRNAs to investigate the effect of genetic variation on killer toxin activity. The large number of killer yeasts with unique antifungal activities discovered since the 1970s suggests that killer toxins are numerous and diverse. Indeed, this is highlighted by the recent description of several novel satellite dsRNAs and associated killer toxins within Saccharomyces yeasts. Ultimately, a better understanding of the relationship between killer toxin genotype and phenotype will afford a better understanding of their contribution to fungal ecology with broad significance to human health and agriculture.”

 Round  2

Reviewer 2 Report

The manuscript has improved considerably since the initial submission and majority of comments were addressed properly. The response to remark on "altering the specificity" of K1 killer (by over-expressing from plasmid) issue differs between one provided in response to reviewer (see last sentence of modified text in response to major comment #1) and corresponding section in updated manuscript. Since the updated manuscript appears to reflect the situation properly, the observed difference is not important anymore.

What remain unresolved though is the issue of observed rather extreme variability of Sc-L-A1 sequence. Authors tend to accept it as intrinsic property of killer system refusing other possibilities, while in the following sentence directly deny their own idea: "It is extremely unlikely that errors in the replication of the dsRNA would result in concurrent mutations
occurring in 20% of all sequenced RT reads." The proximity of frameshift and observed polymorphic regions is an important indication. It is the reverse transcription to be prone for stalling and/or template switching in such context. There are unfortunately no data on sequence reads presented illustrating the link (if any) between direction of reverse transcription and occurrence of sequence polymorphism. Otherwise, it would be very difficult to rationalize the high level of homology between L-A sequences from different viruses. I feel this is an important point to be addressed, since in the present appearance presumed intrinsic sequence variability contradicts data from other researchers, including those generated by NGS.

Author Response

Response to reviewer's comments on January 5th, 2019.

 We have performed additional experiments to investigate the reviewer's concerns regarding a frameshifted region within ScV-L-A1.  We used rt-PCR, cloning, and Sanger sequencing to selectively amplify and sequence the +1 and -1 frameshift.  Based on the results obtained we have downplayed the significance of these mutations identified by NGS and altered the text accordingly: Line 360: “Two polymorphic indels that are present together in 21% of ScV-L-A1 dsRNAs cause a +1 frameshift followed after 55 base pairs by a -1 frameshift, but we were unable to confirm these by reverse transcriptase-PCR, cloning, and Sanger sequencing, meaning that they could have appeared due to replication errors during sample preparation. The proximity of the observed mutations to the secondary structure of the gag-pol frameshift region could account for the observed discrepancy.”

 In addition, references to the 18 aa frameshift have been removed from the discussion and additional methods have been added to describe the approach taken to sequence the ScV-L-A frameshift region. Line 217: “To amplify and clone a putative polymorphic frameshift region from ScV-LA-1, we used reverse transcriptase-PCR with primers PRUI132 and PRUI133. cDNAs were cloned in to into pCR8 by TOPO-TA cloning (Thermo Fisher) and the nucleic acid sequence confirmed by Sanger sequencing.”